# ProxQuant: Quantized Neural Networks via Proximal Operators

**Yu Bai**
Stanford University
yub@stanford.edu

**Yu-Xiang Wang**
UC Santa-Barbara
yuxiangw@cs.ucsb.edu

**Edo Liberty**
Amazon AI
libertye@amazon.com

## Abstract

To make deep neural networks feasible in resource-constrained environments (such as mobile devices), it is beneficial to quantize models by using low-precision weights. One common technique for quantizing neural networks is the straight-through gradient method, which enables back-propagation through the quantization mapping. Despite its empirical success, little is understood about why the straight-through gradient method works.

Building upon a novel observation that the straight-through gradient method is in fact *identical* to Nesterov's dual-averaging algorithm on a quantization constrained optimization problem, we propose a more principled alternative approach, called ProxQuant, that formulates quantized network training as a regularized learning problem instead and optimizes it via the prox-gradient method. ProxQuant does back-propagation on the underlying full-precision vector and applies an efficient prox-operator in between stochastic gradient steps to encourage quantizedness. For quantizing ResNets and LSTMs, ProxQuant outperforms state-of-the-art results on binary quantization and is on par with state-of-the-art on multi-bit quantization. We further perform theoretical analyses showing that ProxQuant converges to stationary points under mild smoothness assumptions, whereas variants such as lazy prox-gradient method can fail to converge in the same setting.

## 1 Introduction

Deep neural networks (DNNs) have achieved impressive results in various machine learning tasks (Goodfellow et al., 2016). High-performance DNNs typically have over tens of layers and millions of parameters, resulting in a high memory usage and a high computational cost at inference time. However, these networks are often desired in environments with limited memory and computational power (such as mobile devices), in which case we would like to compress the network into a smaller, faster network with comparable performance.

A popular way of achieving such compression is through quantization – training networks with low-precision weights and/or activation functions. In a quantized neural network, each weight and/or activation can be representable in $k$ bits, with a possible codebook of negligible additional size compared to the network itself. For example, in a binary neural network ($k = 1$), the weights are restricted to be in $\{\pm 1\}$. Compared with a 32-bit single precision float, a quantized net reduces the memory usage to $k/32$ of a full-precision net with the same architecture (Han et al., 2015; Courbariaux et al., 2015; Rastegari et al., 2016; Hubara et al., 2017; Zhou et al., 2016; Zhu et al., 2016). In addition, the structuredness of the quantized weight matrix can often enable faster matrix-vector product, thereby also accelerating inference (Hubara et al., 2017; Han et al., 2016).

Typically, training a quantized network involves (1) the design of a *quantizer* q that maps a full-precision parameter to a $k$-bit quantized parameter, and (2) the *straight-through gradient method* (Courbariaux et al., 2015) that enables back-propagation from the quantized parameter back onto the original full-precision parameter, which is critical to the success of quantized network training. With quantizer q, an iterate of the straight-through gradient method (see Figure 1a) proceeds

Code available at https://github.com/allenbai01/ProxQuant.

as $\theta_{t+1} = \theta_t - \eta_t \widetilde{\nabla} L(\theta)|_{\theta=\mathsf{q}(\theta_t)}$, and $\mathsf{q}(\widehat{\theta})$ (for the converged $\widehat{\theta}$) is taken as the output model. For training binary networks, choosing $\mathsf{q}(\cdot) = \mathrm{sign}(\cdot)$ gives the BinaryConnect method (Courbariaux et al., 2015).

Though appealingly simple and empirically effective, it is information-theoretically rather mysterious why the straight-through gradient method works well, at least in the binary case: while the goal is to find a parameter $\theta \in \{\pm 1\}^d$ with low loss, the algorithm only has access to stochastic gradients at $\{\pm 1\}^d$. As this is a discrete set, *a priori*, gradients in this set do not necessarily contain any information about the function values. Indeed, a simple one-dimensional example (Figure 1b) shows that BinaryConnect fails to find the minimizer of fairly simple convex Lipschitz functions in $\{\pm 1\}$, due to a lack of gradient information in between.

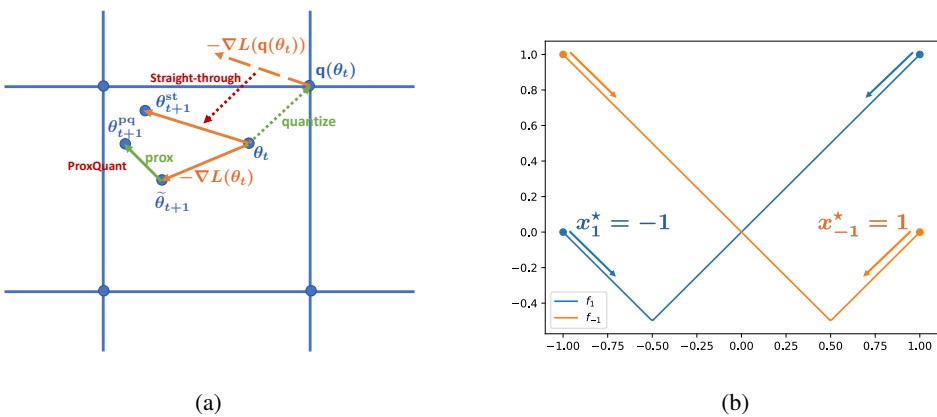

(a)                                                                      (b)

Figure 1: (a) Comparison of the straight-through gradient method and our PROXQUANT method. The straight-through method computes the gradient at the quantized vector and performs the update at the original real vector; PROXQUANT performs a gradient update at the current real vector followed by a prox step which encourages quantizedness. (b) A two-function toy failure case for BinaryConnect. The two functions are $f_1(x) = |x + 0.5| - 0.5$ (blue) and $f_{-1}(x) = |x - 0.5| - 0.5$ (orange). The derivatives of $f_1$ and $f_{-1}$ coincide at $\{-1, 1\}$, so any algorithm that only uses this information will have identical behaviors on these two functions. However, the minimizers in $\{\pm 1\}$ are $x_1^\star = -1$ and $x_{-1}^\star = 1$, so the algorithm must fail on one of them.

In this paper, we formulate the problem of model quantization as a regularized learning problem and propose to solve it with a proximal gradient method. Our contributions are summarized as follows.

- We present a unified framework for defining regularization functionals that encourage binary, ternary, and multi-bit quantized parameters, through penalizing the distance to quantized sets (see Section 3.1). For binary quantization, the resulting regularizer is a $W$-shaped non-smooth regularizer, which shrinks parameters towards either $-1$ or $1$ in the same way that the $L_1$ norm regularization shrinks parameters towards $0$.

- We propose training quantized networks using PROXQUANT (Algorithm 1) — a stochastic proximal gradient method with a homotopy scheme. Compared with the straight-through gradient method, PROXQUANT has access to additional gradient information at non-quantized points, which avoids the problem in Figure 1b and its homotopy scheme prevents potential overshoot early in the training (Section 3.2).

- We demonstrate the effectiveness and flexibility of PROXQUANT through systematic experiments on (1) image classification with ResNets (Section 4.1); (2) language modeling with LSTMs (Section 4.2). The PROXQUANT method outperforms the state-of-the-art results on binary quantization and is comparable with the state-of-the-art on ternary and multi-bit quantization.

- We perform a systematic theoretical study of quantization algorithms, showing that our PROXQUANT (standard prox-gradient method) converges to stataionary points under mild smoothness assumptions (Section 5.1), where as lazy prox-gradient method such as BinaryRelax (Yin et al., 2018) fails to converge in general (Section 5.2). Further, we show that

> BinaryConnect has a very stringent condition to converge to any fixed point (Section 5.3), which we verify through a sign change experiment (Appendix C).

## 1.1 PRIOR WORK

**Methodologies**   Han et al. (2015) propose Deep Compression, which compresses a DNN via sparsification, nearest-neighbor clustering, and Huffman coding. This architecture is then made into a specially designed hardware for efficient inference (Han et al., 2016). In a parallel line of work, Courbariaux et al. (2015) propose BinaryConnect that enables the training of binary neural networks, and Li & Liu (2016); Zhu et al. (2016) extend this method into ternary quantization. Training and inference on quantized nets can be made more efficient by also quantizing the activation (Hubara et al., 2017; Rastegari et al., 2016; Zhou et al., 2016), and such networks have achieved impressive performance on large-scale tasks such as ImageNet classification (Rastegari et al., 2016; Zhu et al., 2016) and object detection (Yin et al., 2016). In the NLP land, quantized language models have been successfully trained using alternating multi-bit quantization (Xu et al., 2018).

**Theories**   Li et al. (2017) prove the convergence rate of stochastic rounding and BinaryConnect on convex problems and demonstrate the advantage of BinaryConnect over stochastic rounding on non-convex problems. Anderson & Berg (2017) demonstrate the effectiveness of binary networks through the observation that the angles between high-dimensional vectors are approximately preserved when binarized, and thus high-quality feature extraction with binary weights is possible. Ding et al. (2018) show a universal approximation theorem for quantized ReLU networks.

**Principled methods**   Sun & Sun (2018) perform model quantization through a Wasserstein regularization term and minimize via the adversarial representation, similar as in Wasserstein GANs (Arjovsky et al., 2017). Their method has the potential of generalizing to other generic requirements on the parameter, but might be hard to tune due to the instability of the inner maximization problem.

Prior to our work, a couple of proximal or regularization based quantization algorithms were proposed as alternatives to the straight-through gradient method, which we now briefly review and compare with. (Yin et al., 2018) propose BinaryRelax, which corresponds to a lazy proximal gradient descent. (Hou et al., 2017; Hou & Kwok, 2018) propose a proximal Newton method with a diagonal approximate Hessian. Carreira-Perpinán (2017); Carreira-Perpinán & Idelbayev (2017) formulate quantized network training as a constrained optimization problem and propose to solve them via augmented Lagrangian methods. Our algorithm is different with all the aforementioned work in using the non-lazy and "soft" proximal gradient descent with a choice of either $\ell_1$ or $\ell_2$ regularization, whose advantage over lazy prox-gradient methods is demonstrated both theoretically (Section 5) and experimentally (Section 4.1 and Appendix C).

## 2 PRELIMINARIES

The optimization difficulty of training quantized models is that they involve a discrete parameter space and hence efficient local-search methods are often prohibitive. For example, the problem of training a binary neural network is to minimize $L(\theta)$ for $\theta \in \{\pm 1\}^d$. Projected SGD on this set will not move unless with an unreasonably large stepsize (Li et al., 2017), whereas greedy nearest-neighbor search requires $d$ forward passes which is intractable for neural networks where $d$ is on the order of millions. Alternatively, quantized training can also be cast as minimizing $L(\mathsf{q}(\theta))$ for $\theta \in \mathbb{R}^d$ and an appropriate *quantizer* $\mathsf{q}$ that maps a real vector to a nearby quantized vector, but $\theta \mapsto \mathsf{q}(\theta)$ is often non-differentiable and piecewise constant (such as the binary case $\mathsf{q}(\cdot) = \mathrm{sign}(\cdot)$), and thus back-propagation through $\mathsf{q}$ does not work.

### 2.1 THE STRAIGHT-THROUGH GRADIENT METHOD

The pioneering work of BinaryConnect (Courbariaux et al., 2015) proposes to solve this problem via the *straight-through gradient method*, that is, propagate the gradient with respect to $\mathsf{q}(\theta)$ unaltered to $\theta$, i.e. to let $\frac{\partial L}{\partial \theta} := \frac{\partial L}{\partial \mathsf{q}(\theta)}$. One iterate of the straight-through gradient method (with the SGD optimizer) is

$$\theta_{t+1} = \theta_t - \eta_t \widetilde{\nabla} L(\theta)|_{\theta = \mathsf{q}(\theta_t)}.$$

This enables the real vector $\theta$ to move in the entire Euclidean space, and taking $\mathsf{q}(\theta)$ at the end of training gives a valid quantized model. Such a customized back-propagation rule yields good empirical performance in training quantized nets and has thus become a standard practice (Courbariaux et al., 2015; Zhu et al., 2016; Xu et al., 2018). However, as we have discussed, it is information theoretically unclear how the straight-through method works, and it does fail on very simple convex Lipschitz functions (Figure 1b).

## 2.2 STRAIGHT-THROUGH GRADIENT AS LAZY PROJECTION

Our first observation is that the straight-through gradient method is equivalent to a *dual-averaging* method, or a lazy projected SGD (Xiao, 2010). In the binary case, we wish to minimize $L(\theta)$ over $\mathcal{Q} = \{\pm 1\}^d$, and the lazy projected SGD proceeds as

$$\begin{cases} \widetilde{\theta}_t = \mathrm{Proj}_{\mathcal{Q}}(\theta_t) = \mathrm{sign}(\theta_t) = \mathsf{q}(\theta_t), \\ \theta_{t+1} = \theta_t - \eta_t \widetilde{\nabla} L(\widetilde{\theta}_t). \end{cases} \tag{1}$$

Written compactly, this is $\theta_{t+1} = \theta_t - \eta_t \widetilde{\nabla} L(\theta)|_{\theta = \mathsf{q}(\theta_t)}$, which is exactly the straight-through gradient method: take the gradient at the quantized vector and perform the update on the original real vector.

## 2.3 PROJECTION AS A LIMITING PROXIMAL OPERATOR

We take a broader point of view that a projection is also a limiting proximal operator with a suitable regularizer, to allow more generality and to motivate our proposed algorithm. Given any set $\mathcal{Q}$, one could identify a regularizer $R : \mathbb{R}^d \to \mathbb{R}_{\geq 0}$ such that the following hold:

$$R(\theta) = 0, \ \forall \theta \in \mathcal{Q} \ \text{ and } \ R(\theta) > 0, \ \forall \theta \notin \mathcal{Q}. \tag{2}$$

In the case $\mathcal{Q} = \{\pm 1\}^d$ for example, one could take

$$R(\theta) = R_{\mathrm{bin}}(\theta) = \sum_{j=1}^{d} \min \{|\theta_j - 1|, |\theta_j + 1|\}. \tag{3}$$

The proximal operator (or prox operator) (Parikh & Boyd, 2014) with respect to $R$ and strength $\lambda > 0$ is

$$\mathrm{prox}_{\lambda R}(\theta) := \underset{\widetilde{\theta} \in \mathbb{R}^d}{\arg\min} \left\{ \frac{1}{2} \left\| \widetilde{\theta} - \theta \right\|_2^2 + \lambda R(\widetilde{\theta}) \right\}.$$

In the limiting case $\lambda = \infty$, the argmin has to satisfy $R(\theta) = 0$, i.e. $\theta \in \mathcal{Q}$, and the prox operator is to minimize $\|\theta - \theta_0\|_2^2$ over $\theta \in \mathcal{Q}$, which is the Euclidean projection onto $\mathcal{Q}$. Hence, projection is also a prox operator with $\lambda = \infty$, and the straight-through gradient estimate is equivalent to a lazy proximal gradient descent with and $\lambda = \infty$.

While the prox operator with $\lambda = \infty$ correponds to "hard" projection onto the discrete set $\mathcal{Q}$, when $\lambda < \infty$ it becomes a "soft" projection that moves towards $\mathcal{Q}$. Compared with the hard projection, a finite $\lambda$ is less aggressive and has the potential advantage of avoiding overshoot early in training. Further, as the prox operator does not strictly enforce quantizedness, it is in principle able to query the gradients at every point in the space, and therefore has access to more information than the straight-through gradient method.

## 3 QUANTIZED NET TRAINING VIA REGULARIZED LEARNING

We propose the PROXQUANT algorithm, which adds a quantization-inducing regularizer onto the loss and optimizes via the (non-lazy) prox-gradient method with a finite $\lambda$. The prototypical version of PROXQUANT is described in Algorithm 1.

---

**Algorithm 1** PROXQUANT: Prox-gradient method for quantized net training

---

**Require:** Regularizer $R$ that induces desired quantizedness, initialization $\theta_0$, learning rates $\{\eta_t\}_{t\geq 0}$, regularization strengths $\{\lambda_t\}_{t\geq 0}$
    **while** not converged **do**
        Perform the prox-gradient step

$$\theta_{t+1} = \text{prox}_{\eta_t \lambda_t R}\left(\theta_t - \eta_t \widetilde{\nabla} L(\theta_t)\right). \tag{4}$$

        The inner SGD step in eq. (4) can be replaced by any preferred stochastic optimization method such as Momentum SGD or Adam (Kingma & Ba, 2014).
    **end while**

---

Compared to usual full-precision training, PROXQUANT only adds a prox step after each stochastic gradient step, hence can be implemented straightforwardly upon existing full-precision training. As the prox step does not need to know how the gradient step is performed, our method adapts to other stochastic optimizers as well such as Adam.

In the remainder of this section, we define a flexible class of quantization-inducing regularizers through "distance to the quantized set", derive efficient algorithms of their corresponding prox operator, and propose a homotopy method for choosing the regularization strengths. Our regularization perspective subsumes most existing algorithms for model-quantization (e.g.,(Courbariaux et al., 2015; Han et al., 2015; Xu et al., 2018)) as limits of certain regularizers with strength $\lambda \rightarrow \infty$. Our proposed method can be viewed as a principled generalization of these methods to $\lambda < \infty$ with a non-lazy prox operator.

## 3.1 REGULARIZATION FOR MODEL QUANTIZATION

Let $\mathcal{Q} \subset \mathbb{R}^d$ be a set of quantized parameter vectors. An ideal regularizer for quantization would be to vanish on $\mathcal{Q}$ and reflect some type of distance to $\mathcal{Q}$ when $\theta \notin \mathcal{Q}$. To achieve this, we propose $L_1$ and $L_2$ regularizers of the form

$$R(\theta) = \inf_{\theta_0 \in \mathcal{Q}} \|\theta - \theta_0\|_1 \quad \text{or} \quad R(\theta) = \inf_{\theta_0 \in \mathcal{Q}} \|\theta - \theta_0\|_2^2. \tag{5}$$

This is a highly flexible framework for designing regularizers, as one could specify any $\mathcal{Q}$ and choose between $L_1$ and $L_2$. Specifically, $\mathcal{Q}$ encodes certain desired quantization structure. By appropriately choosing $\mathcal{Q}$, we can specify which part of the parameter vector to quantize[1], the number of bits to quantize to, whether we allow adaptively-chosen quantization levels and so on. The choice between $\{L_1, L_2\}$ will encourage $\{$"hard","soft"$\}$ quantization respectively, similar as in standard regularized learning (Tibshirani, 1996).

In the following, we present a few examples of regularizers under our framework eq. (5) which induce binary weights, ternary weights and multi-bit quantization. We will also derive efficient algorithms (or approximation heuristics) for solving the prox operators corresponding to these regularizers, which generalize the projection operators used in the straight-through gradient algorithms.

**Binary neural nets** In a binary neural net, the entries of $\theta$ are in $\{\pm 1\}$. A natural choice would be taking $\mathcal{Q} = \{-1, 1\}^d$. The resulting $L_1$ regularizer is

$$R(\theta) = \inf_{\theta_0 \in \{\pm 1\}^d} \|\theta - \theta_0\|_1 = \sum_{j=1}^d \inf_{[\theta_0]_j \in \{\pm 1\}} |\theta_j - [\theta_0]_j|$$

$$= \sum_{j=1}^d \min\{|\theta_j - 1|, |\theta_j + 1|\} = \|\theta - \text{sign}(\theta)\|_1. \tag{6}$$

This is exactly the binary regularizer $R_{\text{bin}}$ that we discussed earlier in eq. (3). Figure 2 plots the W-shaped one-dimensional component of $R_{\text{bin}}$ from which we see its effect for inducing $\{\pm 1\}$ quantization in analog to $L_1$ regularization for inducing exact sparsity.

---

[1]Empirically, it is advantageous to keep the biases of each layers and the BatchNorm layers at full-precision, which is often a negligible fraction, say $1/\sqrt{d}$ of the total number of parameters

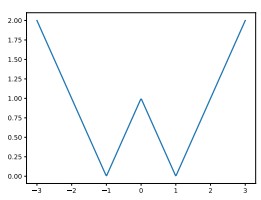

Figure 2: W-shaped regularizer for binary quantization.

The prox operator with respect to $R_{\mathrm{bin}}$, despite being a non-convex optimization problem, admits a simple analytical solution:

$$\mathrm{prox}_{\lambda R_{\mathrm{bin}}}(\theta) = \mathsf{SoftThreshold}(\theta, \mathrm{sign}(\theta), \lambda)$$
$$= \mathrm{sign}(\theta) + \mathrm{sign}(\theta - \mathrm{sign}(\theta)) \odot [|\theta - \mathrm{sign}(\theta)| - \lambda]_+ . \tag{7}$$

We note that the choice of the $L_1$ version is not unique: the squared $L_2$ version works as well, whose prox operator is given by $(\theta + \lambda \,\mathrm{sign}(\theta))/(1+\lambda)$. See Appendix A.1 for the derivation of these prox operators and the definition of the soft thresholding operator.

**Multi-bit quantization with adaptive levels.** Following (Xu et al., 2018), we consider $k$-bit quantized parameters with a structured adaptively-chosen set of quantization levels, which translates into

$$\mathcal{Q} = \left\{ \sum_{i=1}^{k} \alpha_i b_i : \{\alpha_1, \ldots, \alpha_k\} \subset \mathbb{R}, \; b_i \in \{\pm 1\}^d \right\} = \left\{ \theta_0 = B\alpha : \alpha \in \mathbb{R}^k, \; B \in \{\pm 1\}^{d \times k} \right\}. \tag{8}$$

The squared $L_2$ regularizer for this structure is

$$R_{k-\mathrm{bit}}(\theta) = \inf_{\alpha \in \mathbb{R}^k, B \in \{\pm 1\}^{d \times k}} \|\theta - B\alpha\|_2^2, \tag{9}$$

which is also the alternating minimization objective in (Xu et al., 2018).

We now derive the prox operator for the regularizer eq. (9). For any $\theta$, we have

$$\mathrm{prox}_{\lambda R_{k-\mathrm{bit}}}(\theta) = \arg\min_{\widetilde{\theta}} \left\{ \frac{1}{2} \left\| \widetilde{\theta} - \theta \right\|_2^2 + \lambda \inf_{\alpha \in \mathbb{R}^k, B \in \{\pm 1\}^{d \times k}} \left\| \widetilde{\theta} - B\alpha \right\|_2^2 \right\}$$
$$= \arg\min_{\widetilde{\theta}} \inf_{\alpha \in \mathbb{R}^k, B \in \{\pm 1\}^{d \times k}} \left\{ \frac{1}{2} \left\| \widetilde{\theta} - \theta \right\|_2^2 + \lambda \left\| \widetilde{\theta} - B\alpha \right\|_2^2 \right\}. \tag{10}$$

This is a joint minimization problem in $(\widetilde{\theta}, B, \alpha)$, and we adopt an alternating minimization schedule to solve it:

(1) Minimize over $\widetilde{\theta}$ given $(B, \alpha)$, which has a closed-form solution $\widetilde{\theta} = \frac{\theta + 2\lambda B\alpha}{1 + 2\lambda}$.

(2) Minimize over $(B, \alpha)$ given $\widetilde{\theta}$, which does not depend on $\theta_0$, and can be done via calling the alternating quantizer of (Xu et al., 2018): $B\alpha = \mathsf{q}_{\mathrm{alt}}(\widetilde{\theta})$.

Together, the prox operator generalizes the alternating minimization procedure in (Xu et al., 2018), as $\lambda$ governs a trade-off between quantization and closeness to $\theta$. To see that this is a strict generalization, note that for any $\lambda$ the solution of eq. (10) will be an interpolation between the input $\theta$ and its Euclidean projection to $\mathcal{Q}$. As $\lambda \to +\infty$, the prox operator collapses to the projection.

**Ternary quantization** Ternary quantization is a variant of 2-bit quantization, in which weights are constrained to be in $\{-\alpha, 0, \beta\}$ for real values $\alpha, \beta > 0$. We defer the derivation of the ternary prox operator into Appendix A.2.

### 3.2 HOMOTOPY METHOD FOR REGULARIZATION STRENGTH

Recall that the larger $\lambda_t$ is, the more aggressive $\theta_{t+1}$ will move towards the quantized set. An ideal choice would be to (1) force the net to be exactly quantized upon convergence, and (2) not be too aggressive such that the quantized net at convergence is sub-optimal.

We let $\lambda_t$ be a linearly increasing sequence, i.e. $\lambda_t := \lambda \cdot t$ for some hyper-parameter $\lambda > 0$ which we term as the *regularization rate*. With this choice, the stochastic gradient steps will start off close to full-precision training and gradually move towards exact quantizedness, hence the name "homotopy method". The parameter $\lambda$ can be tuned by minimizing the validation loss, and controls the aggressiveness of falling onto the quantization constraint. There is nothing special about the linear increasing scheme, but it is simple enough and works well as we shall see in the experiments.

## 4 EXPERIMENTS

We evaluate the performance of PROXQUANT on two tasks: image classification with ResNets, and language modeling with LSTMs. On both tasks, we show that the default straight-through gradient method is not the only choice, and our PROXQUANT can achieve the same and often better results.

### 4.1 IMAGE CLASSIFICATION ON CIFAR-10

**Problem setup**  We perform image classification on the CIFAR-10 dataset, which contains 50000 training images and 10000 test images of size 32x32. We apply a commonly used data augmentation strategy (pad by 4 pixels on each side, randomly crop to 32x32, do a horizontal flip with probability 0.5, and normalize). Our models are ResNets (He et al., 2016) of depth 20, 32, 44, and 56 with weights quantized to binary or ternary.

**Method**  We use PROXQUANT with regularizer eq. (3) in the binary case and eqs. (15) and (16) in the ternary case, which we respectively denote as PQ-B and PQ-T. We use the homotopy method $\lambda_t = \lambda \cdot t$ with $\lambda = 10^{-4}$ as the regularization strength and Adam with constant learning rate 0.01 as the optimizer.

We compare with BinaryConnect (BC) for binary nets and Trained Ternary Quantization (TTQ) (Zhu et al., 2016) for ternary nets. For BinaryConnect, we train with the recommended Adam optimizer with learning rate decay (Courbariaux et al., 2015) (initial learning rate 0.01, multiply by 0.1 at epoch 81 and 122), which we find leads to the best result for BinaryConnect. For TTQ we compare with the reported results in (Zhu et al., 2016).

For binary quantization, both BC and our PROXQUANT are initialized at the same pre-trained full-precision nets (warm-start) and trained for 300 epochs for fair comparison. For both methods, we perform a hard quantization $\theta \mapsto q(\theta)$ at epoch 200 and keeps training till the 300-th epoch to stabilize the BatchNorm layers. We compare in addition the performance drop relative to full precision nets of BinaryConnect, BinaryRelax (Yin et al., 2018), and our PROXQUANT.

**Result**  The top-1 classification errors for binary quantization are reported in Table 1. Our PROX-QUANT consistently yields better results than BinaryConnect. The performance drop of PROX-QUANT relative to full-precision nets is about $1\%$, better than BinaryConnect by $0.2\%$ on average and significantly better than the reported result of BinaryRelax.

Results and additional details for ternary quantization are deferred to Appendix B.1.

Table 1: Top-1 classification error of binarized ResNets on CIFAR-10. Performance is reported in mean(std) over 4 runs, as well as the (absolute) performance drop of over full-precision nets.

| Model | FP | Classification error | | Performance drop over FP net | | |
| | | BC | PQ-B (ours) | BC | BinaryRelax | PQ-B (ours) |
| (Bits) | (32) | (1) | (1) | (1) | (1) | (1) |
| ResNet-20 | 8.06 | 9.54 (0.03) | **9.35 (0.13)** | +1.48 | +4.84 | **+1.29** |
| ResNet-32 | 7.25 | 8.61 (0.27) | **8.53 (0.15)** | +1.36 | +2.75 | **+1.28** |
| ResNet-44 | 6.96 | 8.23 (0.23) | **7.95 (0.05)** | +1.27 | - | **+0.99** |
| ResNet-56 | 6.54 | 7.97 (0.22) | **7.70 (0.06)** | +1.43 | - | **+1.16** |

### 4.2 LANGUAGE MODELING WITH LSTMS

**Problem setup**  We perform language modeling with LSTMs Hochreiter & Schmidhuber (1997) on the Penn Treebank (PTB) dataset (Marcus et al., 1993), which contains 929K training tokens, 73K validation tokens, and 82K test tokens. Our model is a standard one-hidden-layer LSTM with embedding dimension 300 and hidden dimension 300. We train quantized LSTMs with the encoder, transition matrix, and the decoder quantized to $k$-bits for $k \in \{1, 2, 3\}$. The quantization is performed in a row-wise fashion, so that each row of the matrix has its own codebook $\{\alpha_1, \ldots, \alpha_k\}$.

**Method** We compare our multi-bit PROXQUANT (eq. (10)) to the state-of-the-art alternating minimization algorithm with straight-through gradients (Xu et al., 2018). Training is initialized at a pre-trained full-precision LSTM. We use the SGD optimizer with initial learning rate 20.0 and decay by a factor of 1.2 when the validation error does not improve over an epoch. We train for 80 epochs with batch size 20, BPTT 30, dropout with probability 0.5, and clip the gradient norms to 0.25. The regularization rate $\lambda$ is tuned by finding the best performance on the validation set. In addition to multi-bit quantization, we also report the results for binary LSTMs (weights in $\{\pm 1\}$), comparing BinaryConnect and our PROXQUANT-Binary, where both learning rates are tuned on an exponential grid $\{2.5, 5, 10, 20, 40\}$.

**Result** We report the perplexity-per-word (PPW, lower is better) in Table 2. The performance of PROXQUANT is comparable with the Straight-through gradient method. On Binary LSTMs, PROXQUANT-Binary beats BinaryConnect by a large margin. These results demonstrate that PROXQUANT offers a powerful alternative for training recurrent networks.

Table 2: PPW of quantized LSTM on Penn Treebank.

| Method / Number of Bits | 1 | 2 | 3 | FP (32) |
|---|---|---|---|---|
| BinaryConnect | 372.2 | - | - | |
| PROXQUANT-Binary (ours) | **288.5** | - | - | 88.5 |
| ALT Straight-through[2] | 104.7 | 90.2 | 86.1 | |
| ALT-PROXQUANT (ours) | 106.2 | 90.0 | 87.2 | |

## 5 THEORETICAL ANALYSIS

In this section, we perform a theoretical study on the convergence of quantization algorithms. We show in Section 5.1 that our PROXQUANT algorithm (i.e. non-lazy prox-gradient method) converges under mild smoothness assumptions on the problem. In Section 5.2, we provide a simple example showing that the lazy prox-gradient method fails to converge under the same set of assumptions. In Section 5.3, we show that BinaryConnect has a very stringent condition for converging to a fixed point. Our theory demonstrates the superiority of our proposed PROXQUANT over lazy prox-gradient type algorithms such as BinaryConnect and BinaryRelax (Yin et al., 2018). All missing proofs are deferred to Appendix D.

Prox-gradient algorithms (both lazy and non-lazy) with a fixed $\lambda$ aim to solve the problem

$$\underset{\theta \in \mathbb{R}^d}{\text{minimize}} \, L(\theta) + \lambda R(\theta), \tag{11}$$

and BinaryConnect can be seen as the limiting case of the above with $\lambda = \infty$ (cf. Section 2.2).

### 5.1 A CONVERGENCE THEOREM FOR PROXQUANT

We consider PROXQUANT with batch gradient and constant regularization strength $\lambda_t \equiv \lambda$:

$$\theta_{t+1} = \text{prox}_{\eta_t \lambda R}(\theta_t - \eta_t \nabla L(\theta_t)).$$

**Theorem 5.1** (Convergence of ProxQuant). *Assume that the loss $L$ is $\beta$-smooth (i.e. has $\beta$-Lipschitz gradients) and the regularizer $R$ is differentiable. Let $F_\lambda(\theta) = L(\theta) + \lambda R(\theta)$ be the composite objective and assume that it is bounded below by $F_\star$. Running ProxQuant with batch gradient $\nabla L$, constant stepsize $\eta_t \equiv \eta = \frac{1}{2\beta}$ and $\lambda_t \equiv \lambda$ for $T$ steps, we have the convergence guarantee*

$$\|\nabla F_\lambda(\theta_{T_{\text{best}}})\|_2^2 \leq \frac{C\beta(F_\lambda(\theta_0) - F_\star)}{T} \quad \text{where} \quad T_{\text{best}} = \underset{1 \leq t \leq T}{\arg\min} \|\theta_t - \theta_{t-1}\|_2, \tag{12}$$

*where $C > 0$ is a universal constant.*

---

[2]We thank Xu et al. (2018) for sharing the implementation of this method through a personal communication. There is a very clever trick not mentioned in their paper: after computing the alternating quantization $\mathsf{q}_{\text{alt}}(\theta)$, they multiply by a constant 0.3 before taking the gradient; in other words, their quantizer is a rescaled alternating quantizer: $\theta \mapsto 0.3\mathsf{q}_{\text{alt}}(\theta)$. This scaling step gives a significant gain in performance – without scaling the PPW is $\{116.7, 94.3, 87.3\}$ for $\{1, 2, 3\}$ bits. In contrast, our PROXQUANT does not involve a scaling step and achieves better PPW than this unscaled ALT straight-through method.

**Remark 5.1.** *The convergence guarantee requires both the loss and the regularizer to be smooth. Smoothness of the loss can be satisfied if we use a smooth activation function (such as* tanh*). For the regularizer, the quantization-inducing regularizers defined in Section 3.1 (such as the W-shaped regularizer) are non-differentiable. However, we can use a smoothed version of them that is differentiable and point-wise arbitrarily close to R, which will satisfy the assumptions of Theorem 5.1. The proof of Theorem 5.1 is deferred to Appendix D.1.*

## 5.2 NON-CONVERGENCE OF LAZY PROX-GRADIENT

The lazy prox-gradient algorithm (e.g. BinaryRelax (Yin et al., 2018)) for solving problem eq. (11) is a variant where the gradients are taken at proximal points but accumulated at the original sequence:

$$\theta_{t+1} = \theta_t - \eta_t \nabla L(\text{prox}_{\lambda R}(\theta_t)). \tag{13}$$

Convergence of the lazy prox-gradient algorithm eq. (13) is only known to hold for convex problems (Xiao, 2010); on smooth non-convex problems it generally does not converge even in an ergodic sense. We provide a concrete example that satisfies the assumptions in Theorem 5.1 (so that PROXQUANT converges ergodically) but lazy prox-gradient does not converge.

**Theorem 5.2** (Non-convergence of lazy prox-gradient)**.** *There exists $L$ and $R$ satisfying the assumptions of Theorem 5.1 such that for any constant stepsize $\eta_t \equiv \eta \leq \frac{1}{2\beta}$, there exists some specific initialization $\theta_0$ on which the lazy prox-gradient algorithm eq.* (13) *oscillates between two non-staionry points and hence does not converge in the ergodic sense of eq.* (12)*.*

**Remark 5.2.** *Our construction is a fairly simple example in one-dimension and not very adversarial: $L(\theta) = \frac{1}{2}\theta^2$ and $R$ is a smoothed W-shaped regularizer. See Appendix D.2 for the details.*

## 5.3 CONVERGENCE CHARACTERIZATION FOR BINARYCONNECT

For BinaryConnect, the concept of stationry points is no longer sensible (as the target points $\{\pm 1\}^d$ are isolated and hence every point is stationary). Here, we consider the alternative definition of convergence as converging to a fixed point and show that BinaryConnect has a very stringent convergence condition.

Consider the BinaryConnect method with batch gradients:

$$s_t = \text{sign}(\theta_t), \quad \theta_{t+1} = \theta_t - \eta_t \nabla L(s_t). \tag{14}$$

**Definition 5.1** (Fixed point and convergence)**.** *We say that $s \in \{\pm 1\}^d$ is a **fixed point** of the BinaryConnect algorithm, if $s_0 = s$ in eq.* (14) *implies that $s_t = s$ for all $t = 1, 2, ....$ We say that the BinaryConnect algorithm **converges** if there exists $t < \infty$ such that $s_t$ is a fixed point.*

**Theorem 5.3.** *Assume that the learning rates satisfy $\sum_{t=0}^{\infty} \eta_t = \infty$, then $s \in \{\pm 1\}^d$ is a fixed point for BinaryConnect eq.* (14) *if and only if $\text{sign}(\nabla L(s)[i]) = -s[i]$ for all $i \in [d]$ such that $\nabla L(\theta)[i] \neq 0$. Such a point may not exist, in which case BinaryConnect does not converge for any initialization $\theta_0 \in \mathbb{R}^d$.*

**Remark 5.3.** *Theorem 5.3 is in appearingly a stark contrast with the convergence result for BinaryConnect in (Li et al., 2017) in the convex case, whose bound involves a an additive error $O(\Delta)$ that does not vanish over iterations, where $\Delta$ is the grid size for quantization. Hence, their result is only useful when $\Delta$ is small. In contrast, we consider the original BinaryConnect with $\Delta = 1$, in which case the error makes Li et al. (2017)'s bound vacuous. The proof of Theorem 5.3 is deferred to Appendix D.3.*

**Experimental evidence** We have already seen that such a fixed point $s$ might not exist in the toy example in Figure 1b. In Appendix C, we perform a sign change experiment on CIFAR-10, showing that BinaryConnect indeed fails to converge to a fixed sign pattern, corroborating Theorem 5.3.

## 6 CONCLUSION

In this paper, we propose and experiment with the PROXQUANT method for training quantized networks. Our results demonstrate that PROXQUANT offers a powerful alternative to the straight-through gradient method and has theoretically better convergence properties. For future work, it would be of interest to propose alternative regularizers for ternary and multi-bit PROXQUANT and experiment with our method on larger tasks.

## ACKNOWLEDGEMENT

We thank Tong He, Yifei Ma, Zachary Lipton, and John Duchi for their valuable feedback. We thank Chen Xu and Zhouchen Lin for the insightful discussion on multi-bit quantization and sharing the implementation of (Xu et al., 2018) with us. We thank Ju Sun for sharing the draft of (Sun & Sun, 2018) and the inspiring discussions on adversarial regularization for quantization. The majority of this work was performed when YB and YW were at Amazon AI.

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

## A   ADDITIONAL RESULTS ON REGULARIZATION

### A.1   PROX OPERATORS FOR BINARY NETS

Here we derive the prox operators for the binary regularizer eq. (6) and its squared $L_2$ variant. Recall that

$$R_{\mathrm{bin}}(\theta) = \sum_{j=1}^{d} \min\left\{|\theta_j - 1|, |\theta_j + 1|\right\}.$$

By definition of the prox operator, we have for any $\theta \in \mathbb{R}^d$ that

$$\mathrm{prox}_{\lambda R_{\mathrm{bin}}}(\theta) = \operatorname*{arg\,min}_{\widetilde{\theta} \in \mathbb{R}^d} \left\{ \frac{1}{2} \left\| \widetilde{\theta} - \theta \right\|_2^2 + \lambda \sum_{j=1}^{d} \min\left\{ |\widetilde{\theta}_j - 1|, |\widetilde{\theta}_j + 1| \right\} \right\}$$

$$= \operatorname*{arg\,min}_{\widetilde{\theta} \in \mathbb{R}^d} \left\{ \sum_{j=1}^{d} \frac{1}{2} (\widetilde{\theta}_j - \theta_j)^2 + \lambda \min\left\{ |\widetilde{\theta}_j - 1|, |\widetilde{\theta}_j + 1| \right\} \right\}.$$

This minimization problem is coordinate-wise separable. For each $\widetilde{\theta}_j$, the penalty term remains the same upon flipping the sign, but the quadratic term is smaller when $\mathrm{sign}(\widetilde{\theta}_j) = \mathrm{sign}(\theta_j)$. Hence, the solution $\theta^\star$ to the prox satisfies that $\mathrm{sign}(\theta_j^\star) = \mathrm{sign}(\theta_j)$, and the absolute value satisfies

$$|\theta_j^\star| = \arg\min_{t \geq 0} \left\{ \frac{1}{2}(t - |\theta_j|)^2 + \lambda|t - 1| \right\} = \mathsf{SoftThreshold}(|\theta_j|, 1, \lambda) = 1 + \mathrm{sign}(|\theta_j| - 1)[||\theta_j| - 1| - \lambda]_+.$$

Multiplying by $\mathrm{sign}(\theta_j^\star) = \mathrm{sign}(\theta_j)$, we have

$$\theta_j^\star = \mathsf{SoftThreshold}(\theta_j, \mathrm{sign}(\theta_j), \lambda),$$

which gives eq. (7).

For the squared $L_2$ version, by a similar argument, the corresponding regularizer is

$$R_{\mathrm{bin}}(\theta) = \sum_{j=1}^{d} \min\left\{ (\theta_j - 1)^2, (\theta_j + 1)^2 \right\}.$$

For this regularizer we have

$$\mathrm{prox}_{\lambda R_{\mathrm{bin}}}(\theta) = \arg\min_{\widetilde{\theta} \in \mathbb{R}^d} \left\{ \sum_{j=1}^{d} \frac{1}{2}(\widetilde{\theta}_j - \theta_j)^2 + \lambda \min\left\{ (\widetilde{\theta}_j - 1)^2, (\widetilde{\theta}_j + 1)^2 \right\} \right\}.$$

Using the same argument as in the $L_1$ case, the solution $\theta^\star$ satisfies $\mathrm{sign}(\theta_j^\star) = \mathrm{sign}(\theta_j)$, and

$$|\theta_j^\star| = \arg\min_{t \geq 0} \left\{ \frac{1}{2}(t - |\theta_j|)^2 + \lambda(t - 1)^2 \right\} = \frac{|\theta_j| + \lambda}{1 + \lambda}.$$

Multiplying by $\mathrm{sign}(\theta_j^\star) = \mathrm{sign}(\theta_j)$ gives

$$\theta_j^\star = \frac{\theta_j + \lambda \mathrm{sign}(\theta_j)}{1 + \lambda},$$

or, in vector form, $\theta^\star = (\theta + \lambda \mathrm{sign}(\theta))/(1 + \lambda)$.

## A.2 PROX OPERATOR FOR TERNARY QUANTIZATION

For ternary quantization, we use an approximate version of the alternating prox operator eq. (10): compute $\widetilde{\theta} = \mathrm{prox}_{\lambda R}(\theta)$ by initializing at $\widetilde{\theta} = \theta$ and repeating

$$\widehat{\theta} = \mathsf{q}(\widetilde{\theta}) \quad \text{and} \quad \widetilde{\theta} = \frac{\theta + 2\lambda\widehat{\theta}}{1 + 2\lambda}, \tag{15}$$

where $\mathsf{q}$ is the ternary quantizer defined as

$$\mathsf{q}(\theta) = \theta^+ \mathbf{1}\{\theta \geq \Delta\} + \theta^- \mathbf{1}\{\theta \leq -\Delta\}, \quad \Delta = \frac{0.7}{d}\|\theta\|_1, \quad \theta^+ = \overline{\theta|_{i:\theta_i \geq \Delta}}, \quad \theta^- = \overline{\theta|_{i:\theta_i \leq -\Delta}}. \tag{16}$$

This is a straightforward extension of the TWN quantizer (Li & Liu, 2016) that allows different levels for positives and negatives. We find that two rounds of alternating computation in eq. (15) achieves a good performance, which we use in our experiments.

# B ADDITIONAL EXPERIMENTAL RESULTS

## B.1 TERNARY QUANTIZATION FOR CIFAR-10

Our models are ResNets of depth 20, 32, and 44. Ternarized training is initialized at pre-trained full-precision nets. We perform a hard quantization $\theta \mapsto \mathsf{q}(\theta)$ at epoch 400 and keeps training till the600-th epoch to stabilize the BatchNorm layers.

Table 3: Top-1 classification error of ternarized ResNets on CIFAR-10. Performance is reported in mean(std) over 4 runs, where for PQ-T we report in addition the best of 4 (Bo4).

| Model | FP | TTQ | PQ-T (ours) | PQ-T (ours, Bo4) |
| (Bits) | (32) | (2) | (2) | (2) |
|---|---|---|---|---|
| ResNet-20 | 8.06 | 8.87 | **8.40** (0.13) | **8.22** |
| ResNet-32 | 7.25 | 7.63 | 7.65 (0.15) | **7.53** |
| ResNet-44 | 6.96 | 7.02 | 7.05 (0.08) | **6.98** |

**Result** The top-1 classification errors for ternary quantization are reported in Table 3. Our results are comparable with the reported results of TTQ,[3] and the best performance of our method over 4 runs (from the same initialization) is slightly better than TTQ.

## C  SIGN CHANGE EXPERIMENT

We experimentally compare the training dynamics of PROXQUANT-Binary and BinaryConnect through the *sign change* metric. The sign change metric between any $\theta_1$ and $\theta_2$ is the proportion of their different signs, i.e. the (rescaled) Hamming distance:

$$\mathsf{SignChange}(\theta_1, \theta_2) = \frac{\|\mathrm{sign}(\theta_1) - \mathrm{sign}(\theta_2)\|_1}{2d} \in [0, 1].$$

In $\mathbb{R}^d$, the space of all full-precision parameters, the sign change is a natural distance metric that represents the closeness of the binarization of two parameters.

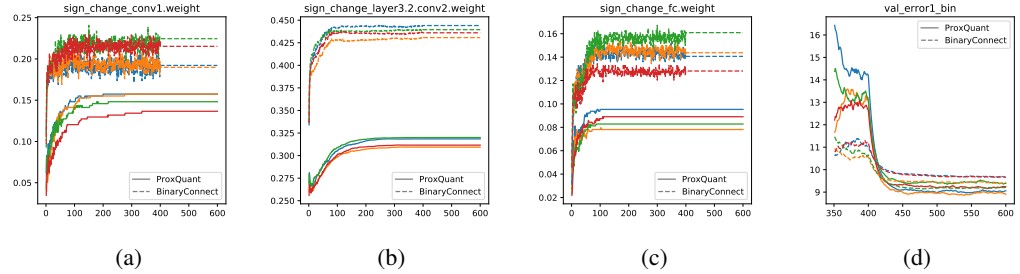

(a)        (b)        (c)        (d)

Figure 3: $\mathsf{SignChange}(\theta_0, \theta_t)$ against $t$ (epoch) for BinaryConnect and PROXQUANT, over 4 runs starting from the same full-precision ResNet-20. PROXQUANT has significantly lower sign changes than BinaryConnect while converging to better models. (a) The first conv layer of size $16 \times 3 \times 3 \times 3$; (b) The last conv layer of size $64 \times 64 \times 3 \times 3$; (c) The fully connected layer of size $64 \times 10$; (d) The validation top-1 error of the binarized nets (with moving average smoothing).

Recall in our CIFAR-10 experiments (Section 4.1), for both BinaryConnect and PROXQUANT, we initialize at a good full-precision net $\theta_0$ and stop at a converged binary network $\widehat{\theta} \in \{\pm 1\}^d$. We are interested in $\mathsf{SignChange}(\theta_0, \theta_t)$ along the training path, as well as $\mathsf{SignChange}(\theta_0, \widehat{\theta})$, i.e. the distance of the final output model to the initialization.

Our finding is that PROXQUANT produces binary nets with both *lower* sign changes and *higher* performances, compared with BinaryConnect. Put differently, around the warm start, there is a good binary net nearby which can be found by PROXQUANT but not BinaryConnect, suggesting that BinaryConnect, and in general the straight-through gradient method, suffers from higher optimization instability than PROXQUANT. This finding is consistent in all layers, across different warm starts, and across different runs from each same warm start (see Figure 3 and Table 4 in Appendix C.1). This result here is also consistent with Theorem 5.3: the signs in BinaryConnect never stop changing until we manually freeze the signs at epoch 400.

---

[3]We note that our PROXQUANT-Ternary and TTQ are not strictly comparable: we have the advantage of using better initializations; TTQ has the advantage of a stronger quantizer: they train the quantization levels $(\theta^+, \theta^-)$ whereas our quantizer eq. (16) pre-computes them from the current full-precision parameter.

## C.1 RAW DATA FOR SIGN CHANGE EXPERIMENT

Table 4: Performances and sign changes on ResNet-20 in mean(std) over 3 full-precision initializations and 4 runs per (initialization x method). Sign changes are computed over all quantized parameters in the net.

| Initialization | Method | Top-1 Error(%) | Sign change |
|---|---|---|---|
| FP-Net 1 | BC | 9.489 (0.223) | 0.383 (0.006) |
| (8.06) | PQ-B | **9.146** (0.212) | **0.276** (0.020) |
| FP-Net 2 | BC | 9.745 (0.422) | 0.381 (0.004) |
| (8.31) | PQ-B | **9.444** (0.067) | **0.288** (0.002) |
| FP-Net 3 | BC | 9.383 (0.211) | 0.359 (0.001) |
| (7.73) | PQ-B | **9.084** (0.241) | **0.275** (0.001) |

Table 5: Performances and sign changes on ResNet-20 in raw data over 3 full-precision initializations and 4 runs per (initialization x method). Sign changes are computed over all quantized parameters in the net.

| Initialization | Method | Top-1 Error(%) | Sign change |
|---|---|---|---|
| FP-Net 1 | BC | 9.664, 9.430, 9.198, 9.663 | 0.386, 0.377, 0.390, 0.381 |
| (8.06) | PQ-B | 9.058, 8.901, 9.388, 9.237 | 0.288, 0.247, 0.284, 0.285 |
| FP-Net 2 | BC | 9.456, 9.530, 9.623, 10.370 | 0.376, 0.379, 0.382, 0.386 |
| (8.31) | PQ-B | 9.522, 9.474, 9.410, 9.370 | 0.291, 0.287, 0.289, 0.287 |
| FP-Net 3 | BC | 9.107, 9.558, 9.538, 9.328 | 0.360, 0.357, 0.359, 0.360 |
| (7.73) | PQ-B | 9.284, 8.866, 9.301, 8.884 | 0.275, 0.276, 0.276, 0.275 |

# D    PROOFS OF THEORETICAL RESULTS

## D.1    PROOF OF THEOREM 5.1

Recall that a function $f : \mathbb{R}^d \to \mathbb{R}$ is said to be $\beta$-smooth if it is differentiable and $\nabla f$ is $\beta$-Lipschitz: for all $x, y \in \mathbb{R}^d$ we have

$$\|\nabla f(x) - \nabla f(y)\|_2 \le \beta \|x - y\|_2.$$

For any $\beta$-smooth function, it satisfies the bound

$$f(y) \le f(x) + \langle \nabla f(x), y - x \rangle + \frac{\beta}{2} \|x - y\|_2^2 \quad \text{for all } x, y \in \mathbb{R}^d.$$

Convergence results like Theorem 5.1 are standard in the literature of proximal algorithms, where we have convergence to stataionarity without convexity on either $L$ or $R$ but assuming smoothness. For completeness we provide a proof below. Note that though the convergence is ergodic, the best index $T_{\text{best}}$ can be obtained in practice via monitoring the proximity $\|\theta_t - \theta_{t-1}\|_2$.

**Proof of Theorem 5.1**    Recall the ProxQuant iterate

$$\theta_{t+1} = \underset{\theta \in \mathbb{R}^d}{\arg\min} \left\{ L(\theta_t) + \langle \theta - \theta_t, \nabla L(\theta_t) \rangle + \frac{1}{2\eta} \|\theta - \theta_t\|_2^2 + \lambda R(\theta) \right\}.$$

By the fact that $\theta_{t+1}$ minimizes the above objective and applying the smoothness of $L$, we get that

$$F_\lambda(\theta_t) = L(\theta_t) + \lambda R(\theta_t) \ge L(\theta_t) + \langle \theta_{t+1} - \theta_t, \nabla L(\theta_t) \rangle + \frac{1}{2\eta} \|\theta_{t+1} - \theta_t\|_2^2 + \lambda R(\theta_{t+1})$$

$$\ge L(\theta_{t+1}) + \left( \frac{1}{2\eta} - \frac{\beta}{2} \right) \|\theta_{t+1} - \theta_t\|_2^2 + \lambda R(\theta_{t+1}) = F_\lambda(\theta_{t+1}) + \frac{\beta}{2} \|\theta_{t+1} - \theta_t\|_2^2.$$

Telescoping the above bound for $t = 0, \dots, T - 1$, we get that

$$\sum_{t=0}^{T-1} \|\theta_{t+1} - \theta_t\|_2^2 \le \frac{2(F_\lambda(\theta_0) - F_\lambda(\theta_T))}{\beta} \le \frac{2(F_\lambda(\theta_0) - F_\star)}{\beta}.$$

Therefore we have the proximity guarantee

$$\min_{0 \le t \le T-1} \|\theta_{t+1} - \theta_t\|_2^2 \le \frac{2(F_\lambda(\theta_0) - F_\star)}{\beta T}. \tag{17}$$

We now turn this into a stationarity guarantee. The first-order optimality condition for $\theta_{t+1}$ gives

$$\nabla L(\theta_t) + \frac{1}{\eta}(\theta_{t+1} - \theta_t) + \lambda \nabla R(\theta_{t+1}) = 0.$$

Combining the above equality and the smoothness of $L$, we get

$$\|\nabla F_\lambda(\theta_{t+1})\|_2 = \|\nabla L(\theta_{t+1}) + \lambda \nabla R(\theta_{t+1})\|_2 = \left\| \frac{1}{\eta}(\theta_t - \theta_{t+1}) + \nabla L(\theta_{t+1}) - \nabla L(\theta_t) \right\|_2$$

$$\le \left( \frac{1}{\eta} + \beta \right) \|\theta_{t+1} - \theta_t\|_2 = 3\beta \|\theta_{t+1} - \theta_t\|_2.$$

Choosing $t = T_{\text{best}} - 1$ and applying the proximity guarantee eq. (17), we get

$$\|\nabla F_\lambda(\theta_{T_{\text{best}}})\|_2^2 \le 9\beta^2 \|\theta_{T_{\text{best}}} - \theta_{T_{\text{best}}-1}\|_2^2 = 9\beta^2 \min_{0 \le t \le T-1} \|\theta_{t+1} - \theta_t\|_2^2 \le \frac{18\beta(F_\lambda(\theta_0) - F_\star)}{T}.$$

This is the desired bound. $\qquad\square$

## D.2    PROOF OF THEOREM 5.2

Let our loss function $L : \mathbb{R} \to \mathbb{R}$ be the quadratic $L(\theta) = \frac{1}{2}\theta^2$ (so that $L$ is $\beta$-smooth with $\beta = 1$). Let the regularizer $R : \mathbb{R} \to \mathbb{R}$ be a smoothed version of the W-shaped regularizer in eq. (3), defined as (for $\epsilon \in (0, 1/2]$ being the smoothing radius)

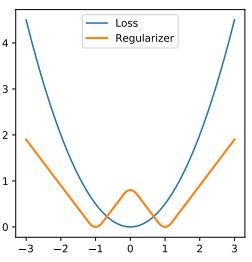

$$R(\theta) = \begin{cases} -\dfrac{1}{2\epsilon}\theta^2 + 1 - \epsilon, & \theta \in [0, \epsilon) \\[2mm] -\theta + 1 - \dfrac{\epsilon}{2}, & \theta \in [\epsilon, 1 - \epsilon) \\[2mm] \dfrac{1}{2\epsilon}(\theta - 1)^2, & \theta \in [1 - \epsilon, 1 + \epsilon) \\[2mm] \theta - 1 - \dfrac{\epsilon}{2}, & \theta \in [1 + \epsilon, \infty) \end{cases}$$

Figure 4

and $R(-\theta) = R(\theta)$ for the negative part. See Figure 4 for an illustration of the loss $L$ and the regularizer $R$ (with $\epsilon = 0.2$).

It is straightforward to see that $R$ is piecewise quadratic and differentiable on $\mathbb{R}$ by computing the derivatives at $\epsilon$ and $1 \pm \epsilon$. Further, by elementary calculus, we can evaluate the prox operator in closed form: for all $\lambda \geq 1$, we have

$$\mathrm{prox}_{\lambda R}(\theta) = \frac{\epsilon\theta + \lambda \, \mathrm{sign}(\theta)}{\epsilon + \lambda \, \mathrm{sign}(\theta)} \quad \text{for all } |\theta| \leq 1.$$

Now, suppose we run the lazy prox-gradient method with constant stepsize $\eta_t \equiv \eta \leq \frac{1}{2\beta} = \frac{1}{2}$. For the specific initialization

$$\theta_0 = \frac{\eta\lambda}{2\lambda + (2 - \eta)\epsilon} \in (0, 1),$$

we have the equality $\mathrm{prox}_{\lambda R}(\theta_0) = \frac{2}{\eta}\theta_0$ and therefore the next lazy prox-gradient iterate is

$$\theta_1 = \theta_0 - \eta \nabla L(\mathrm{prox}_{\lambda R}(\theta_0)) = \theta_0 - \eta \nabla L\left(\frac{2}{\eta}\theta_0\right) = \theta_0 - \eta \cdot \frac{2}{\eta}\theta_0 = -\theta_0.$$

As both $R$ and $L$ are even functions, a symmetric argument holds for $\theta_1$ from which we get $\theta_2 = -\theta_1 = \theta_0$. Therefore the lazy prox-gradient method ends up oscillating between two points:

$$\theta_t = (-1)^t \theta_0.$$

On the other hand, it is straightforward to check that the only stationary points of $L(\theta) + \lambda R(\theta)$ are $0$ and $\pm\frac{\lambda}{\epsilon + \lambda}$, all not equal to $\pm\theta_0$. Therefore the sequence $\{\theta_t\}_{t \geq 0}$ does not have a subsequence with vanishing gradient and thus does not approach stationarity in the ergodic sense. $\qquad\square$

### D.3 Proof of Theorem 5.3

We start with the "$\Rightarrow$" direction. If $s$ is a fixed point, then by definition there exists $\theta_0 \in \mathbb{R}^d$ such that $\theta_t = \theta$ for all $t = 0, 1, 2, \ldots$. By the iterates eq. (14)

$$\theta_T = \theta_0 - \sum_{t=0}^{T} \eta_t \nabla L(s_t).$$

Take signs on both sides and apply $s_t = s$ for all $t$ on both sides, we get that

$$s = s_T = \mathrm{sign}(\theta_T) = \mathrm{sign}\left(\theta_0 - \nabla L(s) \sum_{t=0}^{T} \eta_t\right)$$

Take the limit $T \to \infty$ and apply the assumption that $\sum_t \eta_t = \infty$, we get that for all $i \in [d]$ such that $[\nabla L(\theta)]_i \neq 0$,

$$s[i] = \lim_{T \to \infty} \mathrm{sign}\left(\theta_0 - \nabla L(s) \sum_{t=0}^{T} \eta_t\right)[i] = -\mathrm{sign}(\nabla L(s))[i].$$

Now we prove the "$\Leftarrow$" direction. If $\theta$ obeys that $\text{sign}(\nabla L(s)[i]) = -s[i]$ for all $i \in [d]$ such that $\nabla L(s)[i] \neq 0$, then if we take any $\theta_0$ such that $\text{sign}(\theta_0) = s$, $\theta_t$ will move in a straight line towards the direction of $-\nabla L(s)$, which does not change the sign of $h_0$. In other words, $s_t = \text{sign}(\theta_t) = \text{sign}(\theta_0) = s$ for all $t = 0, 1, 2, ....$ Therefore, by definition, $s$ is a fixed point.

