# OpenReview forum: "ProxQuant: Quantized Neural Networks via Proximal Operators"
_ICLR.cc/2019/Conference_

### Official Review · AnonReviewer3 · 2018-10-25
**Interesting idea but novelty may not be enough**

**Rating:** 5
**Confidence:** 4

**Review:**

After the rebuttal:

1.  Still, the novelty is limited. The authors want to tell a more motivated storyline from Nestrove-dual-average, but that does not contribute to the novelty of this paper. The real difference to the existing works is "using soft instead of hard constraint" for BNN.

2. The convergence is a decoration. It is easy to be obtained from existing convergence proof of proximal gradient algorithms, e.g. [accelerated proximal gradient methods for nonconvex programming. NIPS. 2015].

---------------------------
This paper proposes solving binary nets and it variants using proximal gradient descent. To motivate their method, authors connect lazy projected SGD with straight-through estimator. The connection looks interesting and the paper is well presented. However, the novelty of the submission is limited.

1. My main concern is on the novelty of this paper. While authors find a good story for their method, for example,
- A Proximal Block Coordinate Descent Algorithm for Deep Neural Network Training
- Training Ternary Neural Networks with Exact Proximal Operator
- Loss-aware Binarization of Deep Networks

All above papers are not mentioned in the submission. Thus, from my perspective, the real novelty of this paper is to replace the hard constraint with a soft (penalized) one (section 3.2).

2. Could authors perform experiments with ImageNet?

3. Could authors show the impact of lambda_t on the final performance? e.g., lambda_t = sqrt(t) lambda, lambda_t = sqrt(t^2 lambda

---

> ### Author Response · Authors · 2018-11-12
> **Response**
>
> Thank you for the valuable feedback! We have made a revision to the paper to address all the comments. We will respond to the specific questions in the following.
>
> Novelty --- We agree that there has been a large literature on replacing the straight-through estimator with prox-type algorithms. Our novelty comes in two aspects:
>
> (1) The proposal of combining non-lazy proximal gradient method with a finite (soft) regularization, as well as principled methods for quantizing to binary, ternary, and multi-bit.
>
> (2) A new challenge to the straight-through gradient estimate in its optimization instability through systematic theoretical and empirical investigations. In particular, we show that the convergence criterion of BinaryConnect is very stringent (Theorem 5.1), while our proposed ProxQuant is guaranteed to converge on smooth problems Theorem D.1). Our sign change experiment in Section 5.2 further shows that BinaryConnect is indeed highly unstable in its optimization, as well as giving a lower-performance solution, compared with ProxQuant.
>
> We have updated the related work section (in particular the “Principled methods” part) to include these citations.
>
> ImageNet experiments --- Due to time constraints, we didn’t have time to perform ImageNet experiments for this submission. We have experimental results on LSTMs (Section 4.2) to be complementary with the CIFAR-10 results. Performing ImageNet experiments will be of our interest as a future direction.
>
> Experiments with \lambda_t --- We have thought about that, but we chose to use the linear scheme \lambda_t = \lambda * t for simplicity and to demonstrate that a simple choice would work well. We suspect that changing the schemes would not boost the performance by a great deal -- but we would like to test it experimentally. Please stay tuned and we would potentially add that in our next revision.

---

> ### Author Response · Authors · 2018-11-28
> **Response to the updated review**
>
> Thank you for the quick response after the rebuttal. We respond to the added comments in the following.
>
> --- “Novelty is limited”
> First, we would like to clarify that the difference between our method and BNN are two-fold: our method is a {non-lazy, soft} prox-gradient method whereas BNN (BinaryConnect) is {lazy, hard} (see the discussion in Section 2 and 3.) The difference between lazy projection and standard projection is important as well.
>
> Second, our “motivating storyline” is not only the observation that BinaryConnect is Nesterov’s dual-averaging, but also the observation that BinaryConnect suffers from non-convergence on fairly simple toy problems (Figure 1), and that all lazy prox algorithms suffer from a fundamental information-theoretic limit due to the lack of gradient information as well (Figure 1 & Section 2.3). That poses an issue to address.
>
> More importantly, the contribution of our paper is not merely “proposing yet another alternative to the (already abundant) literature of quantization algorithms”, but also “unveiling the properties of all these algorithms with theories (Section 5) and diagnostic experiments (Appendix C)”. We believe that it will indeed be "an interesting addition to the literature", as Reviewer 1 kindly commented.
>
> --- “Convergence is a decoration”
> We wanted to point out that our theoretical results contain 3 parts: (1) convergence of ProxQuant, (2) non-convergence of lazy proximal gradient descent under the same problem assumptions, and (3) a characterization that BinaryConnect is very unstable and unlikely to converge in practice (with corroborating experiment).
>
> The combination of (1) and (2) points out a clear advantage of ProxQuant over the lazy version in theory, which to the best of our knowledge is novel. Further, the counterexample we construct to show (2) is a very natural simple problem on 1d for binary quantization, which suggest a serious potential drawback of all lazy proximal algorithms in quantization applications. We believe that this result will convey an interesting message to the community about the limitation of the lazy projection mechanism.
>
> Re “convergence can be easily obtained”. We didn’t either claim that the convergence is hard. Indeed, before we present the proof in Appendix D.1, we already remarked that such type of convergence is fairly standard in the literature of proximal algorithms. We will add a reference and attribute the asymptotic critical point convergence guarantee to Atouch et al. (2013). We are not sure if our Ghadimi & Lan (2013) style of convergence **rate** result is new either, but given that it is not hard, we are not claiming any credits for that.
>
> The point of presenting the convergence analysis is to have a self-contained discussion with elementary proofs that separates the standard and lazy prox-gradient algorithms for the problem of model quantization. This particular insight, to the best of our knowledge, is new to the current paper.
>
> - Attouch, H., Bolte, J., & Svaiter, B. F. (2013). Convergence of descent methods for semi-algebraic and tame problems: proximal algorithms, forward–backward splitting, and regularized Gauss–Seidel methods. Mathematical Programming, 137(1-2), 91-129.

---

### Official Review · AnonReviewer2 · 2018-11-02
**Limited theoretical contribution and concerns about experiments**

**Rating:** 7
**Confidence:** 4

**Review:**

This paper proposed ProxQuant method to train neural networks with quantized weights. ProxQuant relax the quantization constraint to a continuous regularizer and then solve the optimization problem with proximal gradient method. The authors argues that previous solvers straight through estimator (STE) in BinaryConnect (Courbariaux et al. 2015) may not converge, and the proposed ProxQuant is better.

 I have concerns about both theoretical and experimental contributions

1. The proposed regularizer for relaxing quantized constraint looks similar to BinaryRelax (Yin et al. 2018 BinaryRelax: A Relaxation Approach For Training Deep Neural Networks With Quantized Weights.), which is not cited. I hope the authors can discuss this work and clarify the novelty of the proposed method. One difference I noticed is that BinaryRelax use lazy prox-graident, while the proposed ProxQuant use non-lazy update. It is unclear which one is better.

2. On page 5, the authors claim ‘’Our proposed method can be viewed as … generalization ...’’ in page 5. It seems inaccurate because unlike proposed method, BinaryConnect use lazy prox-gradient.

3. What’s the purpose of equation (4)? I am confused and did not find it explained in the content.

4. The proposed method introduced more hyper-parameters, like the regularizer parameter \lambda, and the epoch to perform hard quantization. In section 4.2, it is indicated that parameter \lambda is tuned on validation set. I have doubts about the fairness comparing with baseline BinaryConnect. Though BC does not have this parameter, we can still tune learning rate.

5. ProxQuant is fine-tuned based on the pre-trained real-value weights. Is BinaryConnect also fine-tuned? For a CIFAR-10 experiments, 600 epochs are a lot for fine-tuning. As a comparison, training real-value weights usually use less than 300 epochs. BinaryConnect can be trained from scratch using same number of epochs. What does it mean to hard-quantize BinaryConnect? The weights are already quantized after projection step in BinaryConnect.

6. The authors claim there are no reported results with ResNets on CIFAR-10 for BinaryConnect, which is not true. (Li et al. 2017 Training Quantized Nets: A Deeper Understanding) report results on ResNet-56, which I encourage authors to compare with.

7. What is the benefit of ProxQuant? Is it faster than BinaryConnect? If yes, please show convergence curves. Does it generate better results? Table 1 and 2 does not look convincing, especially considering the fairness of comparison.
8. How to interpret Theorem 5.1? For example,  Li et al. 2017 show the real-value weights in BinaryConnect can converge for quadratic function, does it contradict with Theorem 5.1?

9. I would suggest authors to rephrase the last two paragraphs of section 5.2. It first states ‘’one needs to travel further to find a better net’’, and then state ProxQuant find good result nearby, which is confusing.

10.  The theoretical benefit of ProxQuant is only intuitively explained, it looks to me there lacks a rigorous proof to show ProxQuant will converge to a solution of the original quantization constrained problem.

11. The draft is about 9 pages, which is longer than expected. Though the paper is well written and I generally enjoyed reading, I would appreciate it if the authors could shorten the content.

My main concerns are novelty of the proposed method, and fairness of experiments.




======================= after rebuttal =======================

I appreciate the authors' efforts and am generally satisfied with the revision. I raised my score.

The authors show advantage of the proposed ProxQuant over previous BinaryConnect and BinaryRelax in both theory and practice. The analysis bring insights into training quantized neural networks and should be welcomed by the community.

However, I still have concerns about novelty and experiments.

- The proposed ProxQuant is similar to BinaryRelax except for non-lazy vs. lazy updates. I personally like the theoretical analysis showing ProxQuant is better, although it is based on smooth assumptions. However, I am quite surprised BinaryRelax is so much worse than ProxQuant and BinaryConnect in practice (table 1). I would encourage the authors to give more unintuitive explanation.

-  The training time is still long, and the experimental setting seems uncommon. I appreciate the authors' efforts on shortening the finetuning time, and provide more parameter tuning.  However, 200 epochs training full precision network and 300 epochs for finetuning is still a long time, consider previous works like BinaryConnect can train from scratch without a full precision warm start. In this long-training setting, the empirical advantage of ProxQuant over baselines is not much (less than 0.3% for cifar-10 in table 1, and comparable with Xu 2018 in table 2).

---

> ### Author Response · Authors · 2018-11-12
> **Revision and Responses**
>
> Thank you for the very concrete and thoughtful feedback! We have found the comments very useful and constructive for revising the paper.
>
> We have made some initial revisions to address the comments -- please find our changes as well as our response to the comments below.
>
> Novelty and Fairness of Experiments
>
> Point 1 -- As you have pointed out, the main algorithmic difference between ours and Yin et al. (2018) is that we use a non-lazy, standard prox-gradient method whereas their BinaryRelax is a lazy prox-gradient.
>
> The further novelty of our paper lies in the new observation that BinaryConnect suffers from more optimization instability, which are both theoretically and empirically justified in our Section 5.
>
> We have addressed Yin et al. (2018) as well as a few other related literature in the Prior Work subsection (within the “Principled Methods” paragraph), comparing them with our work and highlighting our novelty.
>
> Point 5 -- Both BinaryConnect and our ProxQuant are initialized at pre-trained full-precision nets, which are trained with 200 epochs over CIFAR-10.
>
> For quantization, our schedule is essentially 400 epochs training, and the additional 200 epochs after hard quantization is mostly for fine-tuning the BatchNorm layers. Such fine-tuning was found very useful for *both ProxQuant and BinaryConnect*. Indeed, for BinaryConnect, the signed net keeps changing (in a tiny proportion) even at epoch 400, and the BatchNorm layer hesitates around without being optimized towards any fixed binary net. Hard quantizing forces BinaryConnect to stay at a specific binary net, after which the BatchNorm layer can approach this optimal and boosts performance.
>
> We have modified Section 4.1 to clarify this.
>
> Theoretical Results
>
> Point 8 -- Li et al.’s convergence bound involves an additive error O(\Delta) that does not vanish over iterations, where \Delta is the grid size for quantization. Hence, their result is only useful when \Delta is small. In contrast, we consider the original BinaryConnect with \Delta = 1, in which case the error makes Li et al.’s bound vacuous.
>
> We have added a remark after Theorem 5.1 to clarify that.
>
> Point 9 -- We have rephrased the last two paragraphs in Section 5.2 a bit, to first state our finding and then analyze why it shows the power of ProxQuant over BinaryConnect.
>
> Point 10 -- We have added a convergence guarantee for ProxQuant in Appendix D, showing that ProxQuant converges to a stationary point of the regularized loss.
>
> Presentation
>
> Point 2 -- We have added that we are also using the non-lazy prox to highlight our difference from BinaryConnect.
>
> Point 3 -- The Eq (4) was just an expanded formula for the prox-gradient method. As it did not really mean to say anything and the prox operator has been already defined, we have removed it for clarity.
>
> Point 11 -- We would indeed like to shorten the paper. We will do that once we have a better idea of the potential additional materials that we would present. Please stay tuned.
>
> Additional Experiments
>
> Point 4, 6, 7 -- We will work on some additional experiments to address these points. Please stay tuned and we will let you know once it’s done.
>
> For Point 6 -- The baseline classification error of Adam + BinaryConnect on ResNet56 in Li et. al  is 8.10%, whereas we already achieve a better error 7.79% on ResNet44. We suspect this is due to the difference in the initializing FP net.

---

> > ### Comment · AnonReviewer2 · 2018-11-21
> > **Thanks for response, concerns about novelty and fairness**
> >
> > I appreciate the authors' detailed response. My main concerns are still novelty and fairness. I am willing to raise my score after my main concerns are resolved.
> >
> > 1. I understand there are other contributions such as the analysis of BinaryConnect and ProxQuant in the paper. However, I am still worried about the difference with Yin et al. Like in the authors' response, the main difference seems to be the lazy vs. non-lazy update. Which one is better? Could theoretical or empirical analysis be done for the difference?
> >
> > 2. I am still concerned about experiments and would love to see the authors' response. It looks to me fine-tuning 400 epochs (much more than 200 epochs for standard training) is an uncommon setting. Regarding resnet-56 in Li et al. , it is more about relative number. Li et al. show that with same number of epochs (about 200 for standard training), BinaryConnect can approximate full precision result within <1% difference. In table 1, ProxQuant is better than BinaryConnect within <0.5% difference, but with 600 epochs for training.
> >
> > Thanks.

---

> > > ### Author Response · Authors · 2018-11-25
> > > **Revised, addressing novelty (comparison with Yin et al.) and fairness**
> > >
> > > Thank you again for the suggestions. We have revised our paper again, adding a new section on theoretical analysis and some new experimental results (details also appearing as a new public comment).
> > >
> > > Addressing the comments:
> > >
> > > (1) Novelty, comparison with Yin et al.: We have shown the advantage of our ProxQuant over their BinaryRelax with both theoretical and empirical evidence.
> > >
> > > Theoretically, we have added a non-convergence result for lazy prox-gradient method (e.g. their BinaryRelax) in Section 5.2, which works under the same setting in which our ProxQuant converges (Section 5.1). Specifically, the counter-example we constructed for the non-convergence result is a fairly natural problem in 1d and not very adversarial: quadratic loss, smoothed W-shaped regularizer. Together, we have a comprehensive comparison over lazy and non-lazy prox-gradient methods and shown the advantage of our non-lazy version (ProxQuant).
> > >
> > > Empirically, we have added a comparison of the performance drops of binarization in Table 1, Section 4.1. Our performance drop on CIFAR-10 is typically 1% - 1.3%, much lower than the reported result (2% - 4%) in Yin et al..
> > >
> > > (2) Fairness: number of epochs, ResNet-56, comparison with Li et al.
> > >
> > > We have re-done our CIFAR-10 binarization experiments with 300 epochs (200 training, 100 BatchNorm stabilization), half of what we had before. Our ProxQuant maintains the advantage over BinaryConnect (see Table 1). We have also done experiments on ResNet-56, on which the classification error of {FP, BinaryConnect, ProxQuant} is {6.54%, 7.97%, 7.70%}.
> > >
> > > Specifically, about the comparison with Li et al.: their reported result on ResNet-56 was 8.10% for FP net and 8.83% for BinaryConnect. Though they achieved the <1% performance drop with BinaryConnect, we suspect that may come from the much inferior initializing full-precision net they used (8.10% compared with our 6.54%), so that binarization will cause a lower performance drop on that particular net. In fact their initializing net is even inferior than our full-precision ResNet-20 (8.06%).
> > >
> > > (3) LSTM experiments
> > > We have done a learning rate tuning on binary LSTMs. Both BinaryConnect and our ProxQuant have improved perplexities (BinaryConnect: 372.2, ProxQuant: 288.5) and ProxQuant is still significantly better than BinaryConnect (see Table 2).

---

### Official Review · AnonReviewer1 · 2018-11-02
**An interesting addition to the (large) literature on methods to learn deep networks with quantized weights.**

**Rating:** 8
**Confidence:** 4

**Review:**

This paper proposes a new approach to learning quantized deep neural networks, which overcome some of the drawbacks of previous methods, namely the lack of understanding of why straight-through gradient works and its optimization instability. The core of the proposal is the use of quantization-encouraging regularization, and the derivation of the corresponding proximity operators. Building on that core, the rest of the approach is reasonably standard, based on stochastic proximal gradient descent, with a homotopy scheme.

The experiments on benchmark datasets provide clear evidence that the proposed method doesn't suffer from the drawbacks of  straight-through gradient, does contributing to the state-of-the-art of this class of methods.

---

> ### Author Response · Authors · 2018-11-12
> **Thank you**
>
> Thank you very much for the valuable feedback!

---

### Public Comment · (anonymous) · 2018-09-30
**A highly related paper not cited**

The authors should have cited the paper by Yin et al., first appeared on arXiv  in Jan 2018: https://arxiv.org/pdf/1801.06313.pdf

1. In section 3.1,  the authors propose to replace the hard constraint that imposes the quantization of weights with, for example, a quadratic penalty/regularizer.  The formula of the proximal operator for the quadratic regularizer is derived, which is a weighted average between the weights to be quantized and and the quantized weights as shown in items (1)&(2) below Eq. (11) on page 6.  These contributions are the same as those in section 2.3 of the earlier paper by Yin et al.. Proposition 2.3 in Yin et al.'s paper provided essentially the same proximal operator formula.

2. The authors observe that BinaryConnect iteration can be nicely expressed by Eq. (1) on page 4. The original BinaryConnect paper did not present it explicitly in this way. Their observation of Eq. (1) is basically the same as Eq. (12) on page 9 in Yin et al.'s paper.

---

> ### Author Response · Authors · 2018-10-02
> **Will cite & Differences**
>
> Thanks for bringing the work by Yin et al. to our attention. We were not aware of this paper and did our work independently. We will carefully address this work in our next revision.
>
> We would like to take this opportunity to point out several major differences between our work and Yin et al.:
>
> (1) While we both arrived at the observation that BinaryConnect has a simple expression (our Eq (1) and Yin et al.’s Eq (12)), Yin et al. did not point out this is exactly the dual-averaging algorithm or the lazy-projected gradient descent with constraint set {-1, 1}^d, which dates back to at least Nesterov (for the convex case):
>
> - Nesterov, Y. (2009). Primal-dual subgradient methods for convex problems. Mathematical programming, 120(1), 221-259.
>
> (2) Our algorithm is in fact *different* from Yin et al.: they used the lazy proximal gradient descent (Eq (10), Yin et al.), whereas we used the standard non-lazy proximal gradient descent (our Eq (5)), which is one step further different from the straight-through gradient method.
>
> (3) We proposed and experimented with (1) non-smooth L1-like regularizers for binary quantization; (2) multi-bit quantization with adaptive levels, both not covered in Yin et al..
>
> (4) Our theoretical insights on BinaryConnect (Figure 1 and Section 5) are novel, and in stark contrast with Yin et al.. Our Theorem 5.1 shows that the actual convergence criterion of BinaryConnect is very stringent. We provide a simple 1-d example of such non-convergence in Figure 1.
>
> Our further experimental evidence (Section 5.2) shows that BinaryConnect indeed fails to converge on CIFAR-10 in every run, demonstrating that the condition in Yin et al.’s convergence theorem are quite unlikely to hold in practice.

---

> > ### Author Response · Authors · 2018-11-13
> > **Added some references in the revision**
> >
> > We have added Yin et al., as well as a couple of other relevant literature, into our related work section.

---

### Author Response · Authors · 2018-11-25
**Paper revised: adding theoretical analyses + new experimental results**

We have made a revision to our paper, adding a section on theoretical analysis, as well as some new experimental results. For convenience to the reviewers and readers, we have temporarily highlighted the changes in red (for updated experiments) and blue (for stuff related to theoretical results).

Details of the changes are summarized as follows. Noticably, we now have both theoretical and empirical evidence of our advantage over Yin et al. as described below.

(1) Theoretical analysis
We added a new section for theoretical analysis (Section 5). Specifically, we show that our ProxQuant converges to stationary points under mild smoothness assumptions on the problem (Section 5.1). In the same setting, lazy prox-gradient method (e.g. BinaryRelax of Yin et al.) fails to converge in general -- we construct a fairly natural example in 1d to show that.

Our previous convergence analysis of BinaryConnect is now Section 5.3, and the corresponding sign change experiment is now in Appendix C.

(2) New experimental results
We have shortened the CIFAR-10 training from 600 epochs to 300 epochs (200 training + 100 BatchNorm layer stabilizing) and re-done the experiments. In this new setting, our ProxQuant maintains the advantage over BinaryConnect. This setting also matches the 300 epoch training setup of Yin et al., and our performance drop (~1% - 1.3%) is significantly lower than the reported results of their BinaryRelax (~2% - 4%). We have also added an experiment on ResNet-56.

Due to space constraints and the added binarization results, we have moved the results of ternarization to Appendix B.

For the LSTM experiment, we performed an additional learning rate tuning for the binarized LSTMs. Improved PPW is seen on both BinaryConnect (419.1 -> 372.2) and ProxQuant (321.8 -> 288.5), and ProxQuant still maintains a significant advantage over BinaryConnect.

---

### Public Comment · (anonymous) · 2018-11-26
**The adaptive scalar?**

Hi authors!

I have a question in regards to the binary quantization performed in the experiments. I am curious about how you choose the binary weights. Are they chosen from {-1,1} or {-\alpha, \alpha} with some adaptive real scalar \alpha>0?

I think the adaptive scalar is important to maintain satisfactory precision, e.g., see Xnor-net. But your convergence analysis seems tied to {-1,1}, not {-\alpha, \alpha}.

Can the authors clarify this? Thank you.

---

### Meta-Review · Area_Chair1 · 2018-12-18
**A novel and promising approach to quantized deep nets**

**Confidence:** 4
**Recommendation:** Accept (Poster)

**Metareview:**

A novel  approach for quantized deep neural nets is proposed,  which is more principled than commonly used  straight-through gradient method. A theoretical analysis of the algorithm's converegence  is presented, and empirical results show advantages of the proposed approach.